# Research on Single-Event Burnout Reinforcement Structure of SiC MOSFET

**DOI:** 10.3390/mi15050642

**Published:** 2024-05-10

**Authors:** Qiulan Liao, Hongxia Liu

**Affiliations:** Key Laboratory for Wide Band Gap Semiconductor Materials and Devices of Education, School of Microelectronics, Xidian University, Xi’an 710071, China; 21111223227@stu.xidian.edu.cn

**Keywords:** power device, SiC MOSFET, split gate, SBD-embedded, single-event effect, irradiation hardening

## Abstract

In this paper, the single-event burnout (SEB) and reinforcement structure of 1200 V SiC MOSFET (SG-SBD-MOSFET) with split gate and Schottky barrier diode (SBD) embedded were studied. The device structure was established using Sentaurus TCAD, and the transient current changes of single-event effect (SEE), SEB threshold voltage, as well as the regularity of electric field peak distribution transfer were studied when heavy ions were incident from different regions of the device. Based on SEE analysis of the new structural device, two reinforcement structure designs for SEB resistance were studied, namely the expansion of the P+ body contact area and the design of a multi-layer N-type interval buffer layer. Firstly, two reinforcement schemes for SEB were analyzed separately, and then comprehensive design and analysis were carried out. The results showed that the SEB threshold voltage of heavy ions incident from the N+ source region was increased by 16% when using the P+ body contact area extension alone; when the device is reinforced with a multi-layer N-type interval buffer layer alone, the SEB threshold voltage increases by 29%; the comprehensive use of the P+ body contact area expansion and a multi-layer N-type interval buffer layer reinforcement increased the SEB threshold voltage by 33%. Overall, the breakdown voltage of the reinforced device decreased from 1632.935 V to 1403.135 V, which can be seen as reducing the remaining redundant voltage to 17%. The device’s performance was not significantly affected.

## 1. Introduction

With the development of SiC technology, SiC MOSFET has become one of the fastest growing power devices [1,2]. In power electronic systems, power MOSFET plays an important role in motor-driven inverters and DC–DC conversion circuits. In converter circuits, a diode is usually used in reverse parallel as a freewheeling diode for freewheeling, which can cause problems such as a volume increase and a parasitic effect [3,4,5], thereby limiting switching the frequency and reducing the power density. In theory, freewheeling diodes can directly use the parasitic PIN diodes inside SiC MOSFET. However, on the one hand, prolonged switching between the first and third quadrants of SiC MOSFET can cause a bipolar degradation effect and reduce device reliability [6,7]. On the other hand, the conduction voltage in the third quadrant of the body diode is higher and the conduction loss is greater. In order to solve the above problems, many studies have integrated SBD with lower conduction voltage into SiC MOSFET as a third-quadrant freewheeling diode [8,9,10,11,12,13].

SiC devices are mainly used in DC–DC converters and inverters. In recent years, some device manufacturers have compared and studied device combination technologies and modules, and analyzed the performance of three-phase inverters using three schemes: all SiC, all Si, and SiC/Si hybrid modules [14]. In this article, the first paragraph introduces research on integrating Schottky diodes into SiC MOSFET, which is another device combination technology.

In 1968, Dr. Peter Glaser from the United States first proposed the concept of the space solar power station [15,16]. Various departments in countries such as the United States and Japan carried out a series of designs, research, and evaluation activities, and introduced various system design concepts. The power transmission and management system in space power plants requires the use of ultra-high voltage and high-power equipment. According to China’s engineering of Chasing the Sun—the development strategy of the space solar power station, it is necessary to conduct research on the application of wide bandgap semiconductor power devices at the semiconductor device level. While improving the performance indicators of individual devices, it is also necessary to develop device combination technology. Therefore, the object of this study is SG-SBD-MOSFET.

## 2. Analysis of Single-Event Effect

### 2.1. Device Structure

The structure of the 1200 V SiC MOSFET device with split gate and SBD embedded studied in this article is shown in Figure 1. The relevant parameters are as follows: the doping concentration and thickness of the N-drift are 5.0 × 10^15^ cm^−3^ and 10 μm, respectively. The doping concentration of the substrate is 1.0 × 10^19^ cm^−3^, and Wjfet representing the width of the JFET region is 3 μm. X representing the width of the half-cell grid covering the JFET region is 0.5 μm, and the N-type gaussian doping peak concentration and thickness of CSL are 6.0 × 10^16^ cm^−3^ and 1.65 μm. The design related to the device structure refers to the monograph “Fundamentals of Silicon Carbide Technology: Growth, Characterization, Devices and Applications” [17].

### 2.2. The Physical Models and Other Simulation Information for the TCAD

In this paper, the physical models used for simulation include the impurity incomplete ionization model, the mobility model, the impact ionization model, the drift diffusion model, the carrier recombination model (SRH and Auger), the bandgap narrowing model, et al. In addition, the heavy ion irradiation model will be used when studying SEE [18].

The diagram of the heavy ion irradiation model is shown in Figure 2.

The generation rate of electron hole pairs caused by heavy ions can be calculated using the following equation [18].
(1)$$G(l,\omega ,t)=G_{LET}(l)R(\omega ,l)T(t)$$

In Equation (1), the density of the generated electron hole pairs is determined *G_LET_*(*l*), which can be controlled by defining the Linear Energy Transfer (LET). The variable corresponding to *LET* in the simulation software is LET_f; *R*(*ω,l*) and *T*(*t*) describe the range and time of the generated electron hole pairs respectively.

The occurrence of SEE in devices is influenced by the LET value; LET is used to describe the energy loss of particles, that is, the average energy loss per unit path length. When the incident particles impact with the SiC atomic lattice, the LET value remains unchanged over a short distance, and the energy loss of the incident particle can generally be characterized by the surface LET value of the device. The different LET energy values at the time of the particle incident correspond to different particles.

*R*(*ω*,*l*) can be defined as either an exponential function or a Gaussian function, with the corresponding equation as follows [18]:(2)$$R(\omega ,l)=exp(-\frac{\omega }{\omega_{t}(l)})$$(3)$$R(\omega ,l)=exp[{\left(-\frac{\omega }{\omega_{t}(l)}\right)}^{2}]$$

In Equations (2) and (3), ω is the radius of the ion trajectory, the characteristic distance is *ω_t_*, and the corresponding variable in the simulation software is W_t_hi_.

*T*(*t*) is a Gaussian function; the corresponding equation is as follows [18].
(4)$$T(t)=\frac{2exp[-(\frac{t-t_{0}}{\sqrt{2}\cdot S_{hi}})]}{\sqrt{2}\cdot S_{hi}\sqrt{\pi }(1+erf(\frac{t_{0}}{\sqrt{2}\cdot S_{hi}}))}$$

In Equation (4), *t*_0_ is the time of ion incidence and *S_hi_* is the characteristic value of the Gaussian function.

When adding a heavy ion model to the physical process of the numerical analysis module named Sentaurus Device in the software, the main parameter variables to be set for the heavy ion irradiation model are energy value (Let_f), incident position (Location), incident direction (Direction), incident depth (Length), characteristic distance (W_t_hi_), time, et al. In SEB simulation, the selection of basic parameters for the generation of incident particles is based on the typical values of SiC MOSFETs. Specifically, the particles are incident vertically into the device, the characteristic distance is 0.05 μm, the depth is that the particles are incident throughout the entire device, the LET is 75 MeV·cm^2^/mg, and the time is set to 10^−11^ s [19]. These parameters are initial values set based on typical values in SiC devices, and can be validated and optimized based on the results of the SEE test in the future.

### 2.3. Principle of Single-Event Effect

The process of SEE is as follows: in a blocked state, a large number of electron-hole pairs are generated by impact ionization after heavy ions are incident into SiC MOSFET. In the electric field formed by the bias voltage of the drain-source, electrons move towards the drain electrode, holes move towards the source electrode, enter the P-well, and then move to the P+ body contact region, which changes the electric field distribution in the N-drift, and a high electric field peak appears at the homojunction formed by the N-drift and substrate. In the action of the high electric field, intense impact ionization occurs after acceleration, further producing extremely high concentrations of electron-hole pairs. When the carrier avalanche doubling effect occurs, it causes a transient large current. The source and drain electrodes form a conductive path.

SEB is mainly affected by the electric field. The electric field provides energy for the impact ionization process between electron-hole and silicon carbide atoms. The electron-hole pairs accelerate the impact ionization process by obtaining kinetic energy through the energy provided by the electric field. The relative velocity of generation and recombination of electron-hole pairs is affected by the magnitude of the electric field. When the electric field is weak and the generation speed of the electron-hole pairs is smaller than the recombination speed, the magnitude of the single-event transient current will gradually decrease to zero over time; in the case of a strong electric field, the generation speed of electron-hole pairs is greater than the recombination speed, the transient current remains at a stable level over time, SEB effect occurs, and when the bias voltage applied to the device causes the SEB effect to occur, the critical voltage is defined as the SEB threshold voltage, at which the device that experiences SEB will permanently fail.

### 2.4. Analysis of Single-Event Burnout Effect at Different Incident Positions

The sensitivity of different positions of the device to incident particles varies. Figure 3 shows three SEE incident positions selected from three different regions of the device, namely position A above the midpoint of the JFET region, position B above the edge of the split gate, and position C above the PN junction formed by the N+source and P-well in the source region.

Figure 4 shows the transient current curves at different incident positions in different regions. According to the experimental and the irradiation requirements in the actual space environment, the LET is not less than 75 MeV·cm^2^/mg, the unit of energy is represented by pC/μm in the simulation, and the conversion formula for LET is 1 pC/μm = 151 MeV/mg/cm^2^; therefore, the LET is generally greater than 0.5 pC/μm [20,21]. During the simulation research, the bias conditions are as follows: LET = 0.5 pC/μm, V_GS_ = 0 V, V_DS_ = 400 V.

From Figure 4, it can be seen that the transient current variation trend is consistent at different incident positions. After heavy ions are incident into the device from different positions, the current increases sharply with time, reaching a transient peak current at around 10 ps. Since the bias voltage does not reach the SEB threshold voltage of the device, the transient current will recover to 0 after a certain period of time. Among them, position B has the greatest variation and is most sensitive to the SEB effect, and the transient current changes most dramatically with time.

Figure 5 shows the transient current curves over time at different bias voltages when incident from position B, which is the most sensitive to SEE. From Figure 5, it can be seen that when the bias voltage reaches 430 V, the drain current reaches its peak at 10 ps and then remains at a stable value, and does not return to the initial state after 10^−7^ s, resulting in device burnout. Therefore, the analysis suggests that the threshold voltage for SEB at position B is approximately 430 V. Subsequently, the SEB threshold voltages at positions A and C were analyzed using the same method. Table 1 lists the SEB threshold voltages at different incident positions.

From Table 1, it can be seen that, although position B is more sensitive, the SEB thresholds’ voltage of position A and position B are approximately the same because the current peak of position A and position B are very close. Overall, when heavy ions are incident from the JFET region near the concentration of the electric field, the SEE of the device is more sensitive and the SEB threshold voltage is lower than when they are incident from the source region near the presence of parasitic BJT.

### 2.5. The Regularity of Electric Field Peak Distribution Transfer

SEB is closely related to the internal electric field of the device. Figure 6 shows the electric field distribution transfer when heavy ions are incident from position B on the condition of the drain-source bias voltage of 450 V and SEB occurring.

From Figure 6, it can be seen that before the heavy ion incident, the electric field intensity at the PN junction formed in the P-well and N-drift is the highest. After the heavy ion irradiation, the electrons and holes generated by impact ionization move in the opposite direction at the drain voltage, the holes gradually accumulate in the JFET and channel regions below the gate, causing an increase in the electric field on the channel surface of the device, and the electric field peak gradually transfers to the channel surface. When the accumulated holes cannot be removed in time, the electric field will couple to the gate oxide layer. At 10~100 ps, the maximum electric field is located in the gate oxide layer, as shown in Figure 6e,f. As the high current inside the device changes, a large amount of charge will be concentrated on the substrate surface, which causes the peak electric field to also transfer to there. After 10^−7^ s, the maximum electric field will transfer to the homogeneous junction formed between the N-drift and the substrate surface.

## 3. Design of SEB-Reinforced Structures

### 3.1. Case1: Expansion of P+ Body Contact Area

N+, P-well, and N-drift in the MOSFET form a parasitic NPN bipolar junction transistor (BJT), corresponding to the emitter, base, and collector region of the BJT. Therefore, the basic structure of SiC MOSFET adopts the P+ body contact area at the source, which means that the emission and base of the BJT are connected and short-circuited, thereby suppressing the parasitic BJT. But when the SEE causes the BJT to turn on, the current amplification effect of the BJT can cause a large current to be generated inside the device. Consequently, one idea to increase the device’s SEB threshold is to further suppress the bipolar amplification effect of the BJT, thereby achieving the goal of reinforcement. In this section, SiC MOSFET is designed against radiation by adjusting the area of P+, mainly by extending the P+ body contact area towards the gate direction. The specific schematic diagram is shown in Figure 7:

In this section, the SEB effect was studied when the P+ body contact area is extended in the a (16.7%), b (33%), and c (50%) directions as shown in Figure 7. The simulation results are shown in Table 2.

When the P+ body contact area changes towards three different positions a, b, and c, the impact on device performance is shown in Table 2. Overall, as the P+ body contact area expands towards positions a, b, and c, the SEB threshold voltage gradually increases, and the static characteristics are almost unaffected at positions a and b. However, when it expands to position c, the third-quadrant conduction voltage of the SBD embedded in SiC MOSFET devices will increase. Therefore, reinforcement is considered without affecting the static characteristics of the device; compared to position a, position b has a higher SEB threshold voltage, which has a better reinforcement effect on the SEB when incident from the source pole, which can serve as an extension of the P+ body contact area to improve the device’s resistance to SEB.

The transient current curves of the extended position b in the P+ body contact area at different biases are shown in Figure 8. Heavy ions are vertically incident into SiC MOSFET from position C of the source in Section 2.4.

### 3.2. Case2: Design of Multi-Layer N-Type Interval Buffer Layer

According to the variation regularity of the maximum electric field of the SEE analyzed in Section 2.5, it can be concluded that when heavy ions are incident into SiC MOSFET, the concentration of carriers generated by impact ionization at drain voltage rapidly increases. In the electric field, due to the aggregation of electrons towards the homojunction formed by substrate/N-drift, this is where the high electric field gradually transfers to. Consequently, a reinforcement approach is to disperse the electric field at the homojunction formed by the substrate/N-drift, and the schematic diagram of the reinforcement structure is shown in Figure 9:

Firstly, the thickness of the buffer layer needs to be considered. In this paper, all the thicknesses of the three buffer layers with the multi-layer N-type are 0.8 μm.

#### 3.2.1. Design of Buffer1

In this paper, the doping concentration of the CSL layer is higher than that of the N-drift, which makes it easier for heavy ion incident devices to generate electron-hole pairs and enter the N-drift through the CSL layer, and then transfer to the homojunction formed by the N-drift/substrate, thereby transferring the high electric field to the homojunction. Therefore, buffer1 is set to form a corresponding potential barrier to slow down the drift speed of electron-hole pairs generated by SEE to the substrate surface. The distance between buffer1 and the CSL layer, as well as the doping concentration of buffer1, have a significant impact on the intrinsic breakdown voltage of the device and whether the subsequent addition of buffer2 and buffer3 can achieve the goal of SEB reinforcement. In this section, the distance between buffer1 and the CSL layer is the thickness of the CSL layer. By setting a fixed doping concentration for buffer2 and buffer3 first, the doping concentration of buffer1 is changed to study the variation regularity of SEB reinforcement.

Table 3 shows the changes in device performance when the doping concentration of buffer1 is changed. It can be seen that the SEB threshold voltage remains almost unchanged as the doping concentration of buffer1 increases, while the breakdown voltage decreases significantly. Therefore, minimizing the doping concentration of buffer1 on the basis of having an SEB reinforcement effect to minimize the impact on the breakdown voltage should be considered. Research has shown that removing buffer1 directly cannot achieve the goal of SEB reinforcement. In this paper, the doping concentration of buffer1 is 1 × 10^15^ cm^−3^; that is, the doping concentration of buffer1 is that of the N-drift superimposed by 1 × 10^15^ cm^−3^, and the doping concentration of the N-drift is 5 × 10^15^ cm^−3^. As a result, the actual doping concentration in the buffer1 is 6 × 10^15^ cm^−3^. There is still a small concentration difference between the N-drift and buffer1, forming a potential barrier, which can achieve the goal of SEB reinforcement.

#### 3.2.2. Optimization Design of Buffer2 and Buffer3

The analysis process found that the doping concentration of buffer2 and buffer3 has a relatively small impact on the breakdown voltage of the device, but has a greater impact on the SEB threshold. It is necessary to design the doping concentrations of buffer2 and buffer3 reasonably to ensure that it prevents the buffer layer from penetrating when heavy ion incidence produces SEE leading to the depletion layer broadening effect occurring inside the device in the electric field. At the same time, it bears some of the high electric field at the substrate/N-drift and disperses the severe impact ionization caused by the high electric field at the substrate/N-drift. Overall, the doping concentrations of buffer2 and buffer3 need to initially meet the requirements of 5.0 × 10^15^ cm^−3^ < Dbuffer2 < Dbuffer3 < 1.0 × 10^19^ cm^−3^.

Table 4 shows the impact of the optimized design analysis of buffer2 and buffer3 with different doping concentrations on the SEB threshold voltage and static characteristics of the device.

Taking into account the impact on the performance of the device, doping concentrations of buffer2 and buffer3 were selected to be 3.0 × 10^17^ cm^−3^ and 6.0 × 10^18^ cm^−3^, respectively, for SEB resistance. The transient current over time after the optimized design of buffer2 and buffer3 is shown in Figure 10. Heavy ions are vertically incident into SiC MOSFET from position A which is incident from the center region of the JFET in Section 2.4.

The electric field distribution in the device at different times is shown in Figure 11. it can be seen that before SEB, the electric field inside the device reaches its maximum at the surface. With the change of time, the electric field peak inside the SiC MOSFET device also changes accordingly. As time goes on, the electric field peak inside the device gradually shifts, and the electric field peak is no longer concentrated at the homogeneous junction formed by the N-/substrate. Instead, it spreads towards the direction of the buffer layer. This is because the electric field on the substrate surface is dispersed by the increased buffer layer, thus increasing the SEB threshold voltage.

The transfer diagram of electric field distribution before or after SEB is shown in Figure 12.

### 3.3. Case3: Device Optimized by Comprehensive Reinforcement

After the design of SEB reinforcement according to the incident positions in different regions, the device structure is synthesized as shown in Figure 13. The main design is to increase the P+ body contact area towards the gate direction according to the position b (33%) determined in Figure 7, and to use a multi-layer N-type interval buffer layer design as shown in Figure 9 in the drift region of the device, with a buffer layer thickness of 0.8 μm. The distance between buffer layer buffer1 and the CSL layer is designed to be the thickness of the CSL layer. The doping concentrations of buffer1, buffer2, and buffer3 are 1.0 × 10^15^ cm^−3^, 3.0 × 10^17^ cm^−3^, and 6.0 × 10^18^ cm^−3^, respectively.

The changes in the basic characteristics of the device after the design of comprehensive reinforcement are shown in Table 5. It can be seen that the SEB threshold voltage of heavy ions which are incident from position C at the source of the device decrease by 0.008%, while the SEB threshold voltage of position A which is at the center of the JFET increases by 33%. At the same time, the breakdown voltage of the device decreases by 14%. However, this part of the voltage reduction can be seen as a reduction in the redundant voltage. Generally, the breakdown voltage maintained a 30% margin on the basis of the required withstand voltage when designing the device, while the reinforced device only has 17% redundant voltage left on the basis of the required withstand voltage.

After the design of comprehensive reinforcement, the SEB threshold voltage when heavy ions are incident from position C is actually lower than that of the SEB threshold voltage when using the P+ body contact area expansion reinforcement design alone. However, the SEB threshold voltage when heavy ions are incident from position A is higher than that of the SEB threshold voltage when using a multi-layer N-type interval buffer layer reinforcement design alone. Below is an analysis of the reasons for this result.

Figure 14 shows the equivalent circuit of different regions after the design of comprehensive reinforcement. For the JFET region, the use of a multi-layer N-type interval buffer layer reinforcement is equivalent to connecting three resistors in series on the basis of the integrated SBD, which can disperse the electric field and greatly improve the SEB threshold voltage when incident from the center position of the JFET region, achieving a remarkable irradiation reinforcement effect.

For BJT composed of N+, P-well, and N-drift within the device, using the design of the P+ body contact region expansion reinforcement is equivalent to adding one resistor to the emitter of the BJT. Increasing the emitter resistance is equivalent to enhancing voltage negative feedback, which is equivalent to reducing the input voltage of the BJT, thus playing a role in radiation reinforcement. After adopting the design of a multi-layer N-type interval buffer layer reinforcement, it is equivalent to connecting three resistors in series with the collector of the BJT. Increasing the collector resistance weakens the negative voltage feedback effect, which is equivalent to increasing the input voltage of the BJT, thereby weakening the device’s radiation resistance; as a result, after adopting the design of a combination of the P+ body contact area expansion and a multi-layer N-type interval buffer layer reinforcement, the reinforcement and suppression effect of BJT is cancelled out.

The SEB threshold voltage when heavy ions are incident from the source region and JFET region is generally significantly different, and the SEB threshold voltage when incident from the most sensitive position in the JFET region is much lower than that when incident from the most sensitive position in the source region. Accordingly, increasing the SEB threshold voltage when incident from the more sensitive position in the JFET is equivalent to improving the radiation resistance of the entire device. Overall, the SEB threshold voltage of the device has increased significantly, and the performance against SEE has also been significantly enhanced.

## 4. Conclusions

Regarding the design of SEB reinforcement based on the study of SEB in the device in Section 2, the first reinforcement idea adopts the reinforcement theory of suppressing BJT conduction, adjusting the design of the P+ body contact area for reinforcement, and analyzing the SEE reinforcement through heavy ions from the source-sensitive incident position C of the device; the second reinforcement approach focuses on the severe impact ionization caused by the electric field at the homogeneity junction formed by the N-drift/substrate; a multi-layer N-type interval buffer layer was designed to disperse the electric field concentration. The SEB reinforcement is analyzed when heavy ions are incident from the sensitive position A in the JFET, thereby improving the SEB threshold voltage. Table 6 summarizes the impact of different reinforcement methods on the reinforcement effect and static characteristics of the device.

The scheme of extending the P+ body contact area separately is used to achieve the goal of SEB reinforcement, in order to suppress the current gain and amplification effect of the BJT inside the device. This method has a significant increase of 16% in the SEB threshold voltage when heavy ions are incident from the source region of the device. However, the lack of improvement in the SEB threshold voltage when heavy ions are incident from the JFET region of the device is not significant. The JFET region is relatively more sensitive, and the SEB threshold voltage is lower, which better characterizes the anti-SEB ability of the entire device. Consequently, this method also needs to be designed in conjunction with other reinforcement methods.

The design of using a multi-layer N-type spacer buffer layer alone for SEB reinforcement of the device effectively transfers the concentrated large electric field from the N-drift/substrate homojunction to the surface of the buffer layer, thereby the SEB threshold voltage has been improved significantly. This method can increase the SEB threshold voltage when heavy ions are incident from the JFET of the device by 29%; it is a relatively effective method for SEB reinforcement. However, the breakdown voltage of the device is reduced. In this method, the breakdown voltage is only affected by the doping concentration of the first buffer1; while the doping concentrations of buffer2 and buffer3 determine the upper limit of the increase in the SEB threshold voltage, the breakdown voltage decreases from the original 1632.935 V to 1392.659 V. For devices with a voltage withstand value of 1200 V, there is already a 30% margin left on the basis of 1200 V in the design, and the reduction in breakdown voltage of the reinforced device can be seen as less redundant voltage left on the basis of the required withstand voltage.

The design which adopts a combination of a P+ body contact area expansion and a multi-layer N-type interval buffer layer cancels the reinforcement of the BJT suppression, but still has a good reinforcement effect on high electric field dispersion. Moreover, compared to the design of using multi-layer N-shaped interval buffer layer reinforcement alone, it can improve the reinforcement effect and the SEB threshold voltage greatly when incident from the center position of the JFET; in other words, the radiation resistance of the entire device can be improved. The SEB threshold voltage is increased by 33%.

## Figures and Tables

**Figure 1 micromachines-15-00642-f001:**
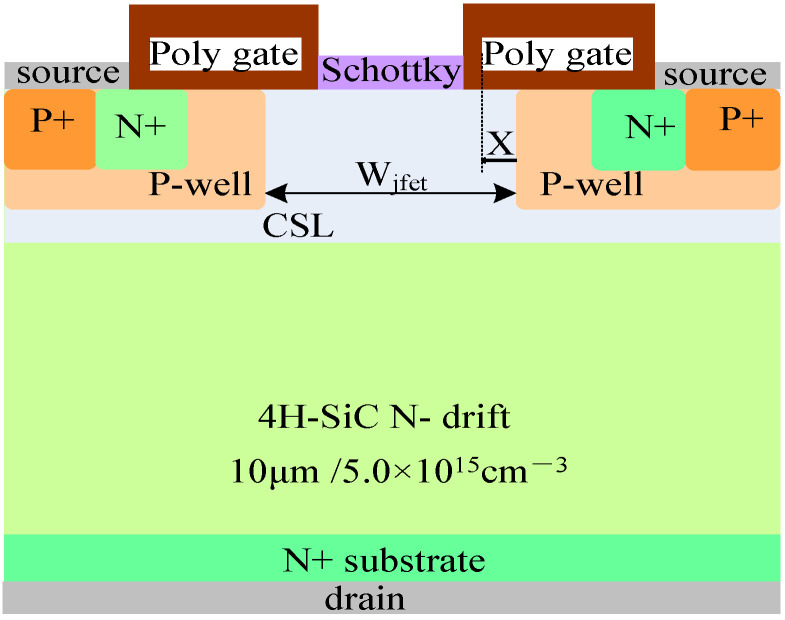
1200 V SiC MOSFET with split gate and SBD embedded.

**Figure 2 micromachines-15-00642-f002:**
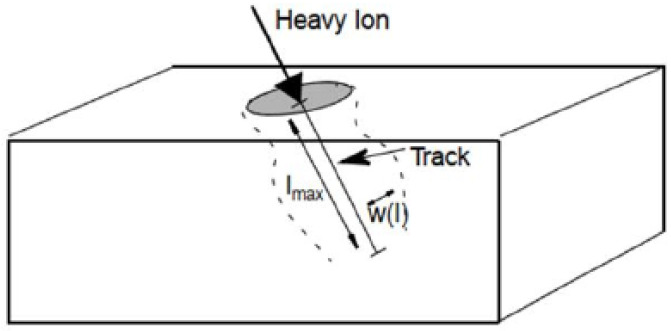
Heavy ion irradiation model [18].

**Figure 3 micromachines-15-00642-f003:**
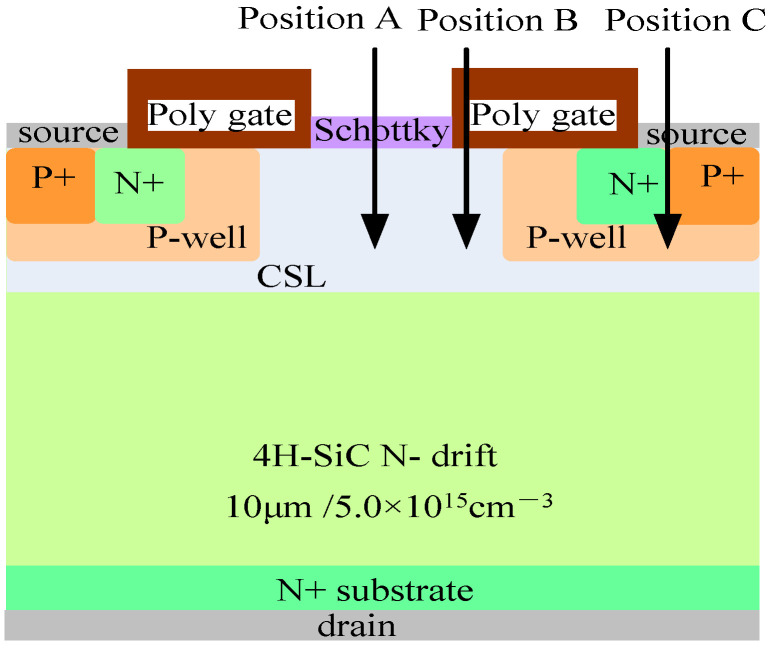
Incident positions in different regions.

**Figure 4 micromachines-15-00642-f004:**
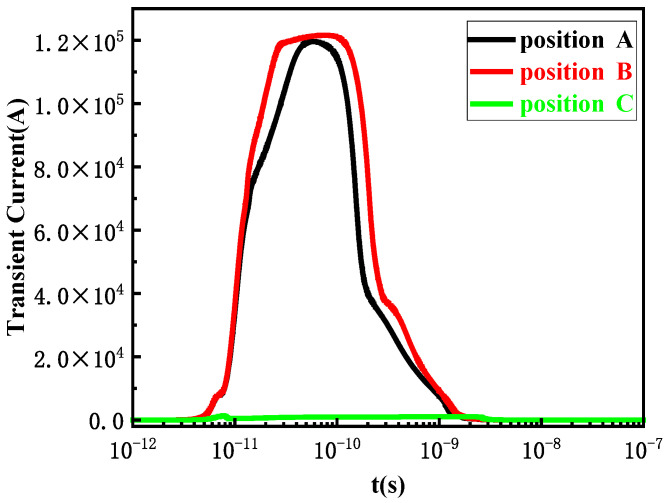
Transient current variation curves at different incident positions in different regions.

**Figure 5 micromachines-15-00642-f005:**
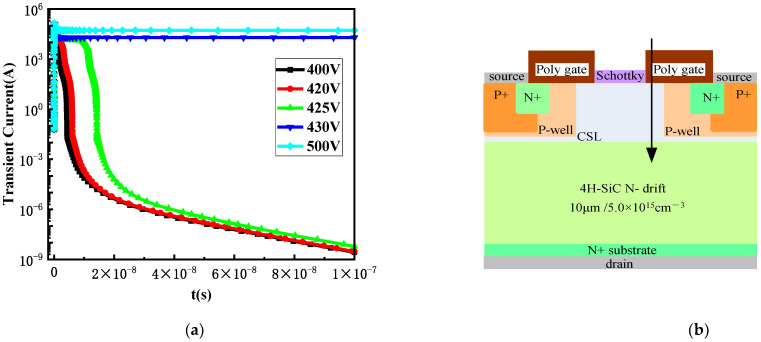
(**a**) Transient current curves of heavy ions incident from position B at different biases; (**b**) incident position B.

**Figure 6 micromachines-15-00642-f006:**
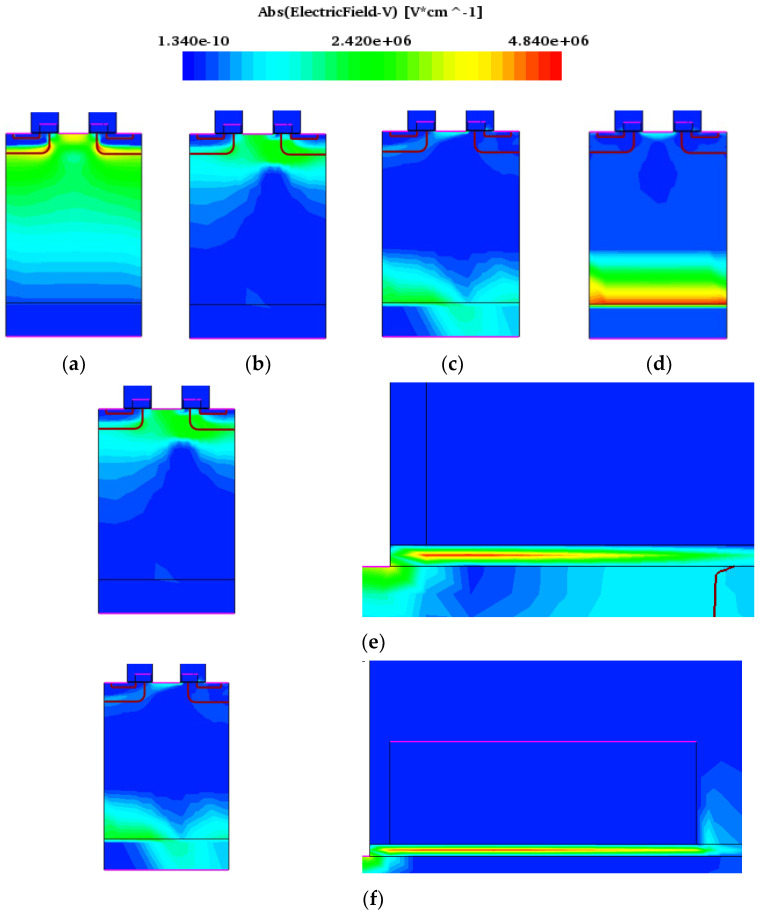
Distribution trend of electric field peak transfer during SEB (V_DS_ = 450 V: (**a**) 1 ps; (**b**) 10 ps; (**c**) 100 ps; (**d**) 10^−7^ s; (**e**) enlarged diagram of electric field distribution in the oxide layer at 10 ps; (**f**) enlarged diagram of electric field distribution in the oxide layer at 100 ps.

**Figure 7 micromachines-15-00642-f007:**
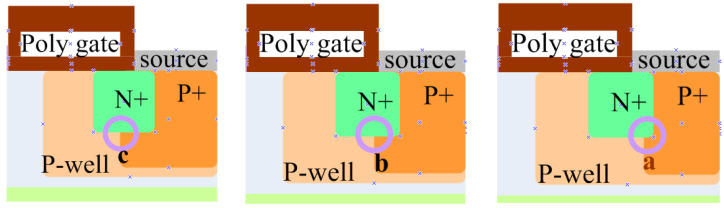
Schematic diagram of the expansion positions of different P+ body contact areas.

**Figure 8 micromachines-15-00642-f008:**
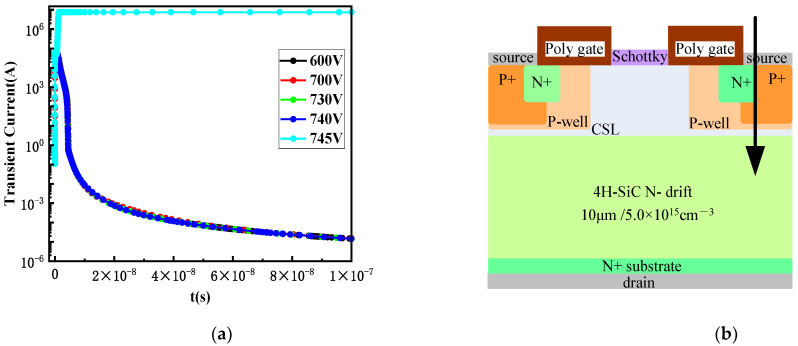
(**a**) Transient current curve of the extended position b in the P+ body contact area; (**b**) incident position C.

**Figure 9 micromachines-15-00642-f009:**
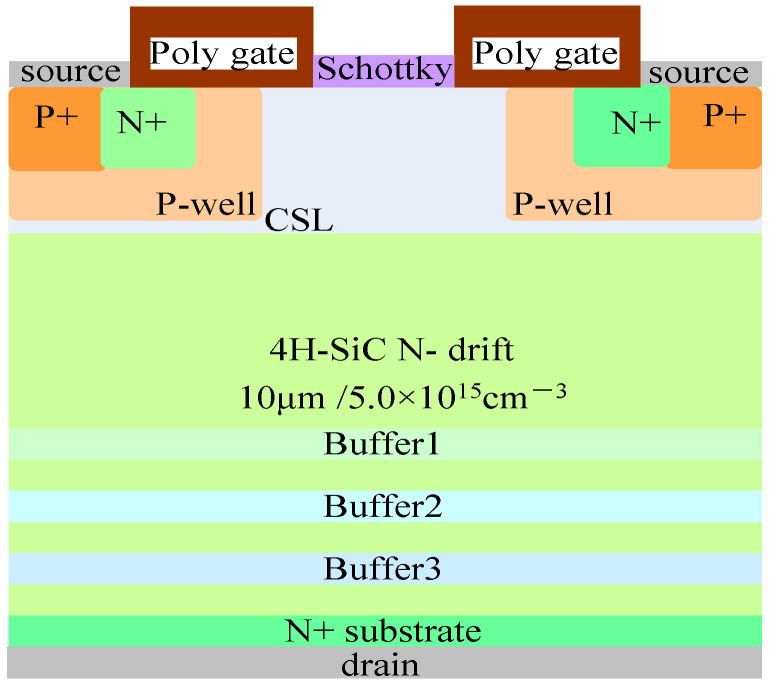
Structure with multi-layer N-type interval buffer layer.

**Figure 10 micromachines-15-00642-f010:**
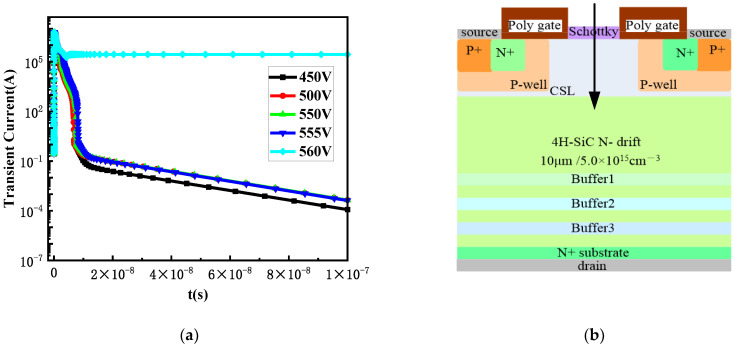
(**a**) Curves at different bias voltages after optimized design of buffer2 and buffer3; (**b**) incident position A.

**Figure 11 micromachines-15-00642-f011:**
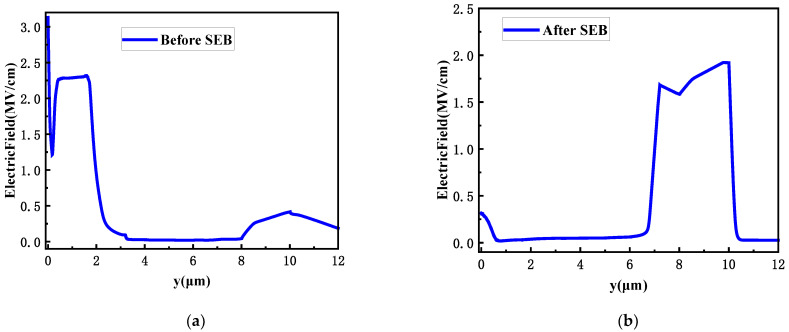
(**a**) Electric field distribution before SEB with buffer layer design; (**b**) Electric field distribution after SEB with buffer layer design.

**Figure 12 micromachines-15-00642-f012:**
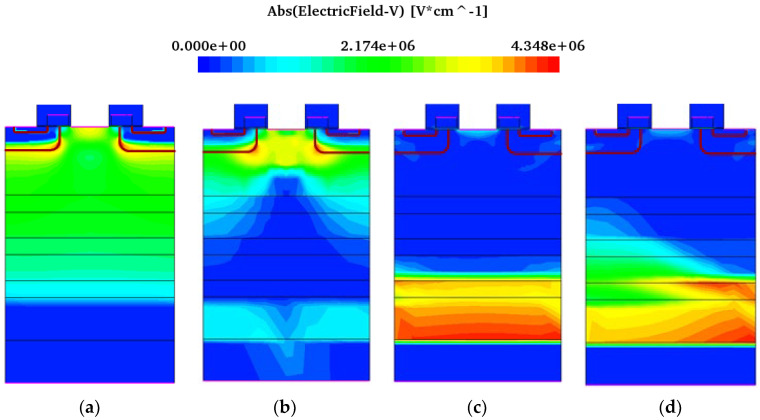
Electric field distribution transfer for SEB with buffer layer design: (**a**) 1 ps; (**b**) 10 ps; (**c**) 1000 ps; (**d**) 10^−7^ s.

**Figure 13 micromachines-15-00642-f013:**
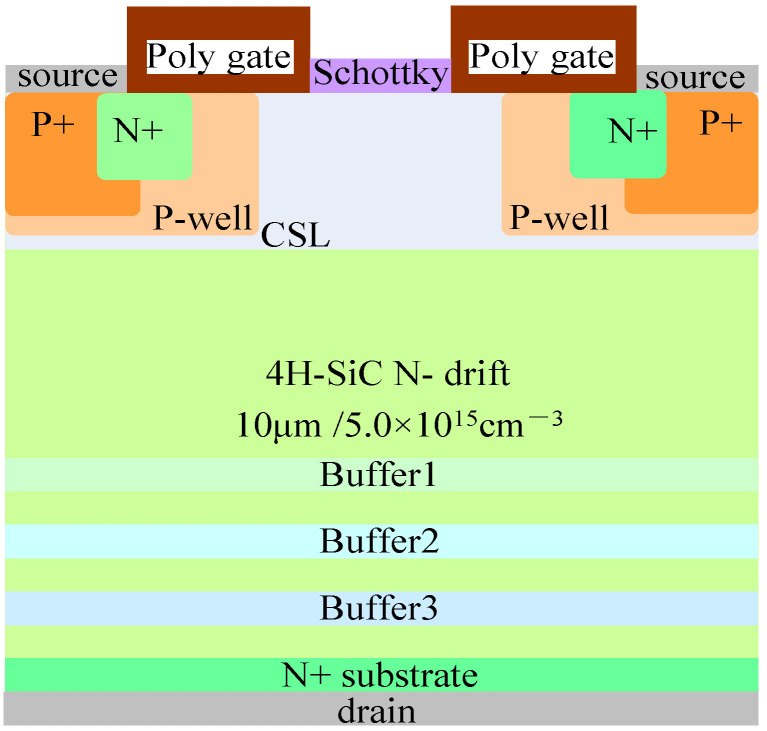
Device structure after comprehensive reinforcement optimization.

**Figure 14 micromachines-15-00642-f014:**
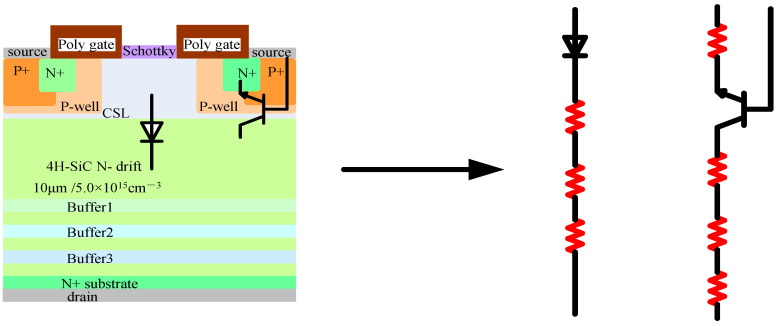
Equivalent circuit in different areas after comprehensive reinforcement.

**Table 1 micromachines-15-00642-t001:** SEB threshold voltage at different incident positions.

Incident Position	Position A	Position B	Position C
SEB threshold voltage (V)	430	430	640

**Table 2 micromachines-15-00642-t002:** Impact of P+ body contact area extension on device performance.

Expand Location	Unexpanded	a	b	c
SEB threshold voltage (V)	640	735	740	745
Third-quadrant conduction voltage (V)	1.283	1.283	1.283	1.312
Breakdown voltage of the device (V)	1632.935	1641.787	1642.935	1642.888

**Table 3 micromachines-15-00642-t003:** Changes in device characteristics after buffer layer reinforcement design.

Reinforcement Design and Device Characteristics	Data					
Doping Concentration of Buffer1 (cm^−3^)	-	1.0 × 10^15^	6.0 × 10^15^	8.0 × 10^15^	1.0 × 10^16^	1.5 × 10^16^
SEB threshold voltage (V)	430	440	440	440	440	440
Breakdown voltage of the device (V)	1632.935	1392.659	1359.178	1342.803	1333.024	1293.024

**Table 4 micromachines-15-00642-t004:** Changes in device characteristics after optimized design of buffer2 and buffer3.

Reinforcement Design and Device Characteristics	Data			
Doping Concentration of Buffer2 (cm^−3^)	1 × 10^17^	1 × 10^17^	3 × 10^17^	3 × 10^17^
Doping concentration of Buffer3 (cm^−3^)	8 × 10^17^	1 × 10^18^	3 × 10^18^	6 × 10^18^
SEB threshold voltage (V)	430	440	505	555
Breakdown voltage of the device (V)	1392.659	1392.659	1392.659	1392.659

**Table 5 micromachines-15-00642-t005:** Changes in characteristics before or after comprehensive reinforcement design.

Before or after Reinforcement	Unreinforced	Case3
SEB threshold voltage for position C (V)	640	635
SEB threshold voltage for position B (V)	430	555
SEB threshold voltage for position A (V)	430	570
Breakdown voltage of the device (V)	1632.935	1403.135

**Table 6 micromachines-15-00642-t006:** Changes in characteristics before or after different reinforcement designs.

Before or after Reinforcement	Unreinforced	Case1	Case2	Case3
SEB threshold voltage for position C (V)	640	740	--	635
SEB threshold voltage for position A (V)	430	--	555	570
Breakdown voltage of the device (V)	1632.935	1642.935	1392.659	1403.135

## Data Availability

The original contributions presented in the study are included in the article, further inquiries can be directed to the corresponding author.

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
