# Peer review of "Research on Single-Event Burnout Reinforcement Structure of SiC MOSFET"

_micromachines, 2024, doi:10.3390/mi15050642_

Round 1

Reviewer 1 Report

Comments and Suggestions for Authors

1. The information of CSL such as material and dopping in Fig. 1 and Fig. 2 should be reported.

2. Position B is the most sensitive area according to Table 1 and 2.4 statement , but this position is not used to obtain the BV of the proopsed structure, how about the reinforcment performance of the expansion of P+body contact area for position B ?

3. The physical models and other simulation information for the TCAD model should be provided.

Author Response

1. Summary

Thank you very much for taking the time to review this manuscript. We appreciate your meticulous and careful review of this manuscript, as well as your constructive suggestions on many details. Please find the detailed responses below and the corresponding revisions/corrections highlighted in the re-submitted files.

2. Point-by-point response to Comments and Suggestions for Authors

Comments 1: The information of CSL such as material and dopping in Fig. 1 and Fig. 2 should be reported

Response 1:Thank you for pointing this out. We have added relevant information, which you can see from lines 65-68 of the updated manuscript.

Comments 2: Position B is the most sensitive area according to Table 1 and 2.4 statement , but this position is not used to obtain the BV of the proopsed structure, how about the reinforcment performance of the expansion of P+body contact area for position B

Response 2: Thank you for your question. As the expansion and reinforcement of the P+body contact area is mainly designed based on the principle of suppressing parasitic BJT, only the SEB threshold voltage reinforcement effect at position C in the source region where parasitic BJT exists is shown. For position B far away from the parasitic BJT region, it hardly changes the SEB threshold voltage. You can see this explanatory information from lines 369-370 in the updated manuscript.

Comments 3: The physical models and other simulation information for the TCAD model should be provided

Response 3: Thank you for your suggestion. We agree with this comment. Therefore, we have made relevant supplements and added a section to provide the model information. You can see this explanatory information in section 2.2 of lines 70-106 of the updated manuscript.

Reviewer 2 Report

Comments and Suggestions for Authors

This manuscript describes a proposed SiC MOSFET design that intends to enhance the single-event burnout (SEB) voltage of the device under radiation. The authors propose three different forms of reinforcement by way of adding additional layers of doped material far away from the active region that traps the electric field during/after the single-event effect (SEE). In general the layout of this manuscript serves well in presenting logical advancements in device design. However, there are many concerns regarding the motivation of the study, the presentation of experiments conducted, and interoperability of this study. 

The study in general serves some merit, but I cannot recommend for publication in the current state. Significant work must be done to better address the needs of the field and to more clearly present the study that has been conducted for other researchers. The discussion surrounding Cases 1-3 constantly change notation and formatting in a way that appears to attempt to hide information from the reader.

1.) The introduction needs citations to reinforce every claim made by the authors. This section serves as an introduction and motivation for the problem that this manuscript is attempting to solve. Please see below for examples:

Line 28: '...SiC MOSFETS have become one of the fastest growing power devices.' should have a citation that shows a baseline example of the interest of the field. A good review article for example would serve excellently here. 

Line 32: '...cause problems such as volume increase and parasitic effect,...' needs to have a citation that shows this is a real problem that the field is currently trying to solve. 

Lines 40-42: This sentence is extremely out of place with no context. 

After reading the introduction, I do not feel like this study has been properly motivated. The work is well put together, but it is unclear why the study is necessary.

2.) This is a minor issue in terms of readability for the paragraph from lines 84-90, but 'LET' is never defined in the text. This can easily be solved with a stronger introduction and motivation that contextualizes this study better.

3.) I don't understand the layouts presented in Figure 5e and f. Specifically, the side profiles have never been presented earlier in the text and it is unclear what is being presented in these figures.

4.) Case 1 presents the findings when increasing the contact area of the P+ body. What is the increase in area studied here? Is it an increase by 5%? 10%? 20%? I understand that Figure 6 presents a schematic of this increase, but the actual value of area increase is necessary. This is also an issue since this contact area (at least by the figures) is constantly changing across Cases 1-3. 

5.) Case 3 presents a "comprehensive reinforcement optimization" device. What does this mean? What parameters from Cases 1 and 2 did you use to create this comprehensive design.

6.) Why was position B omitted from the discussion of Case 3?

7.) The simulation methodologies are never specified in the text. Without a discussion of the assumptions made and comprehensive device parameters it is impossible for anyone else to use this work as it stands.  

Comments on the Quality of English Language

There are a few minor grammatical issues, namely "electricfield" should be "electric field", with a space between "electric" and "field". 

There is a missed space in line 123 after the comma after the word 'device'

The sentence from line 120-124 that contains the semi-colon, the first clause of this sentence does not appear to have a subject and is a fractured sentence.

Author Response

1. Summary

Thank you very much for taking the time to review this manuscript. We appreciate your meticulous and careful review of this manuscript, as well as your constructive suggestions on many details. Please find the detailed responses below and the corresponding revisions/corrections highlighted in the re-submitted files.

2. Point-by-point response to Comments and Suggestions for Authors

Comments 1:The introduction needs citations to reinforce every claim made by the authors. This section serves as an introduction and motivation for the problem that this manuscript is attempting to solve. Please see below for examples:

Line 28: '...SiC MOSFETS have become one of the fastest growing power devices.' should have a citation that shows a baseline example of the interest of the field. A good review article for example would serve excellently here.

Response 1: Thank you for pointing this out. We have added relevant information, which you can see from lines 28-29 of the updated manuscript.

Line 32: '...cause problems such as volume increase and parasitic effect,...' needs to have a citation that shows this is a real problem that the field is currently trying to solve.

Response 1:Thank you for pointing this out. We have added relevant information, which you can see from line 33 of the updated manuscript.

Lines 40-42: This sentence is extremely out of place with no context.

Response 1:Thank you for pointing this out. We have further explained the context, which you can see from lines 41-57 of the updated manuscript.

Comments 2: This is a minor issue in terms of readability for the paragraph from lines 84-90, but 'LET' is never defined in the text. This can easily be solved with a stronger introduction and motivation that contextualizes this study better.

Response 2: Thank you for your question. We have supplemented this information and you can see it in lines 83-88 of the updated manuscript.

Comments 3: I don't understand the layouts presented in Figure 5e and f. Specifically, the side profiles have never been presented earlier in the text and it is unclear what is being presented in these figures.

Response 3: Thank you for your question. The left figure in Figures 5 (e) and (f) represents the overall cross-sectional view of the device at 10ps and 100ps, respectively, while the right figure represents an enlarged view of the gate oxide layer at 10ps and 100ps, respectively. These are explained in the title of the figure (in the updated manuscript, they are Figures 6 (e) and (f)). Overall, it represents that the maximum value of the electric field is transferred to the gate oxide layer at 10ps to 100ps. You can see this explanatory information from lines 183-184 of the updated manuscript.

Comments 4: Case 1 presents the findings when increasing the contact area of the P+ body. What is the increase in area studied here? Is it an increase by 5%? 10%? 20%? I understand that Figure 6 presents a schematic of this increase, but the actual value of area increase is necessary. This is also an issue since this contact area (at least by the figures) is constantly changing across Cases 1-3.

Response 4: Thank you for your suggestion. We have made relevant supplements, and you can see this explanatory information in line 204 of the updated manuscript.

Comments 5: Case 3 presents a "comprehensive reinforcement optimization" device. What does this mean? What parameters from Cases 1 and 2 did you use to create this comprehensive design.

Response 5: Thank you for your question. We have added some explanatory information, which you can see from lines 299-305 of the updated manuscript.

Comments 6: Why was position B omitted from the discussion of Case 3?

Response 6: Thank you for your question. We have added the information for position B in Table 5. In addition, the reinforced design significantly increases the SEB threshold voltage for position A. Therefore, we mainly selected position A. You can see this information in the updated manuscript.

Comments 7: The simulation methodologies are never specified in the text. Without a discussion of the assumptions made and comprehensive device parameters it is impossible for anyone else to use this work as it stands.

Response 7: Thank you for your suggestion. We agree with this comment. Therefore, we have made relevant supplements and added a section to provide the model information. You can see this explanatory information in section 2.2 of lines 60-106 of the updated manuscript.

3. Response to Comments on the Quality of English Language

Point 1:There are a few minor grammatical issues, namely "electricfield" should be "electric field", with a space between "electric" and "field"

Response 1: Thank you for carefully reviewing the manuscript. We have corrected this issue in the updated manuscript

Point 2:There is a missed space in line 123 after the comma after the word 'device.

Response 2: Thank you for carefully reviewing the manuscript. We have corrected this issue in the updated manuscript, you can see this change from line 181 in the updated manuscript.

Point 3:The sentence from line 120-124 that contains the semi-colon, the first clause of this sentence does not appear to have a subject and is a fractured sentence.

Response 3: Thank you for carefully reviewing the manuscript. We have corrected this issue in the manuscript, you can see this change from lines 177-182 in the updated manuscript.

Round 2

Reviewer 1 Report

Comments and Suggestions for Authors

1. When the abbreviation "CSL" first appears, the full name should be provided.

Reviewer 2 Report

Comments and Suggestions for Authors

The authors have revised their original manuscript titled "Research on Single Event Burnout Reinforcement Structure of SiC MOSFET" to include a much more descriptive motivation for study and a significantly enhanced presentation of theoretical and simulation methods. With the edits made, this has become a very nice manuscript and I recommend it for publication.